# Inflammatory stress signaling via NF-*k*B alters accessible cholesterol to upregulate SREBP2 transcriptional activity in endothelial cells

Joseph Wayne M Fowler, Rong Zhang, Bo Tao, Nabil E Boutagy, William C Sessa*

Vascular Biology and Therapeutics Program, Department of Pharmacology, Yale University School of Medicine, New Haven, United States

**Abstract** There is a growing appreciation that a tight relationship exists between cholesterol homeostasis and immunity in leukocytes; however, this relationship has not been deeply explored in the vascular endothelium. Endothelial cells (ECs) rapidly respond to extrinsic signals, such as tissue damage or microbial infection, by upregulating factors to activate and recruit circulating leukocytes to the site of injury and aberrant activation of ECs leads to inflammatory based diseases, such as multiple sclerosis and atherosclerosis. Here, we studied the role of cholesterol and a key transcription regulator of cholesterol homeostasis, SREBP2, in the EC responses to inflammatory stress. Treatment of primary human ECs with pro-inflammatory cytokines upregulated SREBP2 cleavage and cholesterol biosynthetic gene expression within the late phase of the acute inflammatory response. Furthermore, SREBP2 activation was dependent on NF-κB DNA binding and canonical SCAP-SREBP2 processing. Mechanistically, inflammatory activation of SREBP was mediated by a reduction in accessible cholesterol, leading to heightened sterol sensing and downstream SREBP2 cleavage. Detailed analysis of NF-κB inducible genes that may impact sterol sensing resulted in the identification of a novel *RELA*-inducible target, *STARD10*, that mediates accessible cholesterol homeostasis in ECs. Thus, this study provides an in-depth characterization of the relationship between cholesterol homeostasis and the acute inflammatory response in EC.

## Editor's evaluation

This is a fundamental contribution to linking inflammation to cholesterol metabolism in endothelial cells. The strength of evidence was compelling and the uncovering of the molecular mechanisms underlying the pathway was a significant addition to the overall value of the study.

## Introduction

The majority of the biological processes regulating acute inflammation has focused on the contribution of tissue-infiltrating leukocytes. Undoubtedly, leukocytes are crucial for host defense and tissue repair, regulating the balance between resolution and chronic inflammation. However, the endothelium plays a significant role in the overall inflammatory response, particularly in initiation and vascular maintenance. Endothelial cells (ECs) are in constant contact with the bloodstream and rapidly change their phenotype in response to inflammatory stimuli. Inflammatory cytokines, such as tumor necrosis factor alpha (TNFα) and interleukin-1 beta (IL1β), bind to their respective receptors to activate I-κ-kinase, which phosphorylates and degrades inhibitory IκBα and releases the key inflammatory transcription factor, NF-κB, to the nucleus (*DiDonato et al., 1997*). NF-κB, along with other activated

*For correspondence:
william.sessa@yale.edu

Competing interest: The authors declare that no competing interests exist.

transcription factors, such as activator protein 1 (AP1) upregulate the transcription of several inflammatory response genes that increase (1) vascular permeability, (2) leukocyte chemoattraction, and (3) immune cell adhesion and extravasation into tissue (*Pober and Sessa, 2007*). Indeed, the vascular endothelium is a primary sensor of the circulating bloodstream and is exposed to various stimuli that regulate systemic host defense responses.

It is becoming increasingly appreciated that there exists a connection between cellular immunity and cholesterol. Cellular lipid and cholesterol homeostasis are tightly regulated by the transcription factor sterol response element binding protein (SREBP). At sufficient cellular cholesterol levels, SREBP is retained as a full-length protein in the endoplasmic reticulum (ER) bound to adaptor proteins SREBP cleavage-activating protein (SCAP) and inhibitory insulin-induced gene (INSIG) (*Brown and Goldstein, 1997*). When cellular cholesterol levels decrease, the SCAP-SREBP complex translocates to the Golgi where SREBP is proteolytically cleaved by proteases, S1P and S2P. Cleavage results in the release of the N-terminal fragment of SREBP into the cytoplasm, which translocates to the nucleus to bind to DNA and initiate gene transcription. SREBP1a and SREBP1c isoforms predominantly activate the expression of genes involved in fatty acid synthesis and the SREBP2 isoform upregulates genes that increase cellular cholesterol by de novo synthesis and exogenous uptake (*Horton et al., 2002*).

The relationship between SREBP2, cholesterol homeostasis, and immune phenotype has been predominantly studied in leukocyte immunobiology. First, it has been suggested SREBP2 directly modulates immune responses. In macrophages, it was found that the SCAP/SREBP2 shuttling complex directly interacts with the NLRP3 inflammasome and regulates inflammasome activation via translocation from ER to Golgi (*Guo et al., 2018*). Another group found that SREBP2 was highly activated in macrophages treated with TNFα and that nuclear SREBP2 bound to inflammatory and interferon response target genes to promote an M1-like inflammatory state (*Kusnadi et al., 2019*). Second, several studies have shown that cellular cholesterol levels control immune phenotype. Type I interferon (IFN) signaling in macrophages decreases cholesterol synthesis, allowing for activation of STING on the ER to feed forward and enhance IFN signaling (*York et al., 2015*). Furthermore, decreasing cholesterol synthesis via *Srebf2* knockout was sufficient to activate the type I IFN response. Mevalonate, an intermediate in the cholesterol biosynthetic pathway, can regulate trained immunity in monocytes. Patients lacking mevalonate kinase accumulate mevalonate and develop hyper immunoglobulin D syndrome (*Bekkering et al., 2018*). On the other hand, studies have indicated that bacterial lipopolysaccharide (LPS) or type I IFNs can actively suppress synthesis of cholesterol in macrophages and that restoring cholesterol biosynthesis promotes inflammation (*Araldi et al., 2017*; *Dang et al., 2017*).

Despite the importance of the endothelium in the inflammatory response, the link between inflammation and cholesterol homeostasis is not well studied in ECs. There is evidence that increased activation of SREBP2 and cholesterol loading in ECs are pro-inflammatory, but these studies were largely focused on models of atherosclerosis and did not focus on the mechanism of activation, flux of the discrete pools of cholesterol and/or sterol sensing in ECs (*Xiao et al., 2013*; *Westerterp et al., 2016*). Here we show an intimate relationship between inflammatory signaling and cholesterol homeostasis in EC. Tumor necrosis factor α (TNFα) rapidly activates NF-κB resulting in a time-dependent activation of SREBP2 and SREBP2 dependent gene expression. The activation of SREBP occurs via TNF mediated stimulation of NF-κB and changes in the accessible cholesterol pool that promote sterol sensing but not via other post-translational mechanism. Mechanistically, we show that TNFα induction of the NF-κB inducible gene, *STARD10,* in part, mediates the changes in accessible cholesterol leading to heightened SREBP activation. Thus, EC respond to inflammatory cytokine challenges by reducing the accessible cholesterol pool on the plasma membrane thereby inducing canonical SREBP processing and gene expression leading to inflammation.

## Results

### TNFα and RELA transcriptionally regulate canonical SREBP-dependent gene expression

Primary HUVEC were treated with TNFα (10 ng/mL) for 4 and 10 hr followed by RNA-seq analysis to uncover the transcriptional changes at peak and later stages of activation, respectively. As both a positive control and exploratory aim, we performed RNA sequencing on HUVEC treated with TNFα

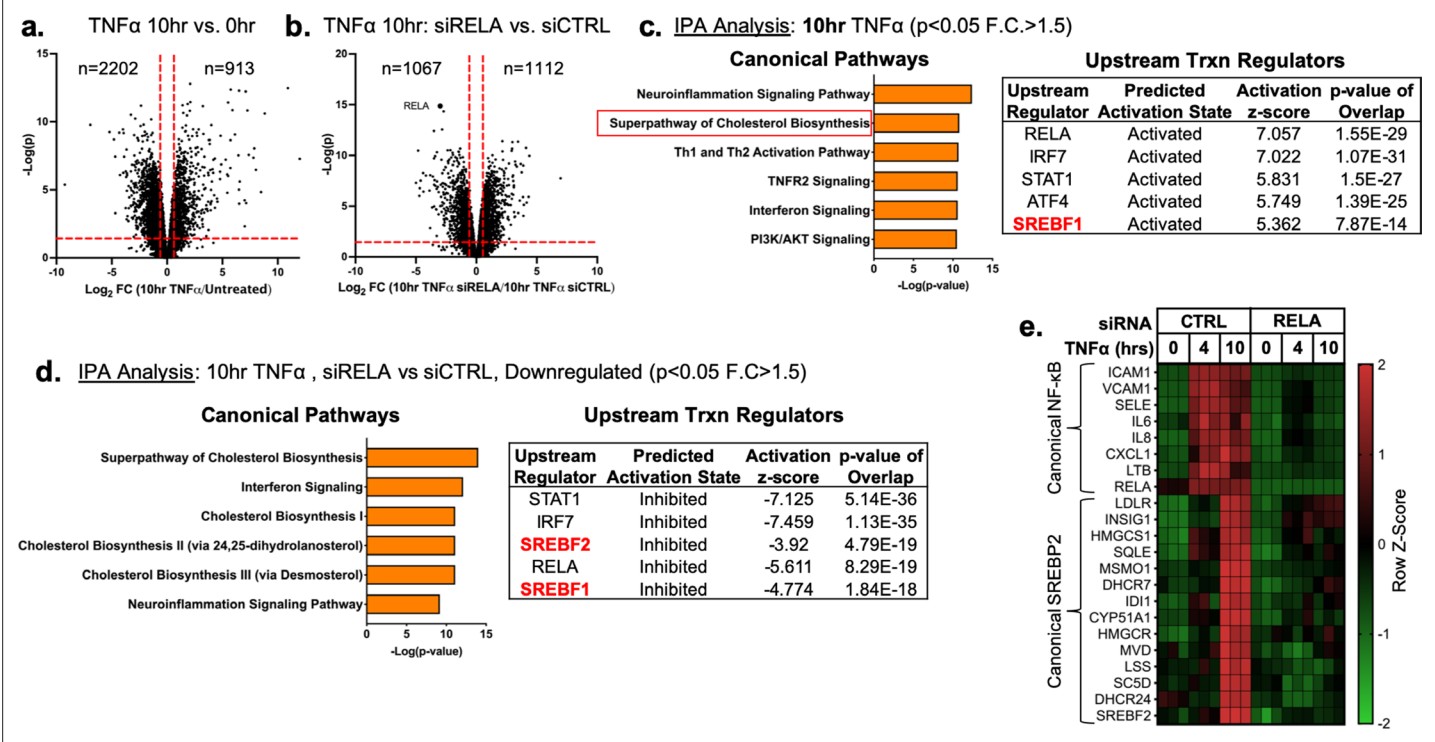

**Figure 1.** TNFα and NF-κB control SREBP2-dependent gene expression in human endothelial cells. Primary HUVEC were treated siRNA against non-targeting sequence (si*CTRL*) or *RELA* for 48 hr and then incubated with or without 10 ng/mL TNFα for 10 hr. (a) Volcano plot for RNA-seq analysis of differentially expressed genes. Dotted red lines indicate cutoff used for IPA analysis (p<0.05, 1.5<Fold Change (F.C)<−1.5). (b) IPA analysis of most significant canonical pathways and predicted upstream transcriptional regulators for genes that increase at 10 hr TNFα. (c) IPA analysis of most significant canonical pathways and predicted upstream transcriptional regulators for genes that decrease in cells knocked down with *RELA* siRNA and treated 10 hr TNFα compared to control cells treated with 10 hr TNFα. (d) Representative heatmap of NF-κB and SREBP2 transcriptionally controlled genes from (b) and (c) showing three independent donors.

The online version of this article includes the following figure supplement(s) for figure 1:

**Figure supplement 1.** Complete transcriptomic pathway analysis in HUVEC treated with TNFa for 0, 4, and 10 hr and with or with *RELA* siRNA.

at similar timepoints after RNAi-mediated knockdown of *RELA*, which encodes the protein P65, the key DNA-binding component of the canonical NFκB transcriptional complex. Treatment of HUVEC for 10 hr with TNFα resulted in significant upregulation of 913 genes and downregulation of 2202 genes (p<0.05; −1.5>Fold Change (F.C)>1.5) (*Figure 1a*) and *RELA* knockdown decreased expression of 1067 genes and increased expression of 1112 genes (*Figure 1b*). As expected, *RELA* gene expression was the most significantly gene decreased after knockdown.

Ingenuity Pathway Analysis (IPA) revealed that TNFα treatment for 4 hr resulted in the upregulation of several expected pathways reported in literature, including inflammation, TNFR signaling, and activation of IRF (*Figure 1—figure supplement 1a*; *Hogan et al., 2017*). These pathways were also significantly upregulated in HUVEC treated after 10 hr of TNFα treatment (*Figure 1c*). Interestingly, Canonical Pathway Analysis uncovered the 'Superpathway of Cholesterol Biosynthesis' as the second most significant pathway upregulated in the 10 hr treatment group. Furthermore, Upstream Regulator analysis restricted to transcription factors predicted that SREBF1 was significantly activated in these cells. Metacore analysis of metabolic networks and GSEA hallmark analysis similarly revealed significant upregulation of the cholesterol homeostasis pathway at the 10 hr timepoint (*Figure 1—figure supplement 1b and c*).

RNA-seq pathway analysis of genes reduced after *RELA* knockdown in HUVEC treated with TNFα (10 hr) revealed that expected inflammatory pathways, such as interferon signaling and neuroinflammation, were significantly inhibited when RELA was not present (*Figure 1d*). Additionally, the 'Superpathway of Cholesterol Biosynthesis' and several other redundant pathways populated the most significant Canonical Pathway results. Upstream transcription regulator analysis predicted SREBP2

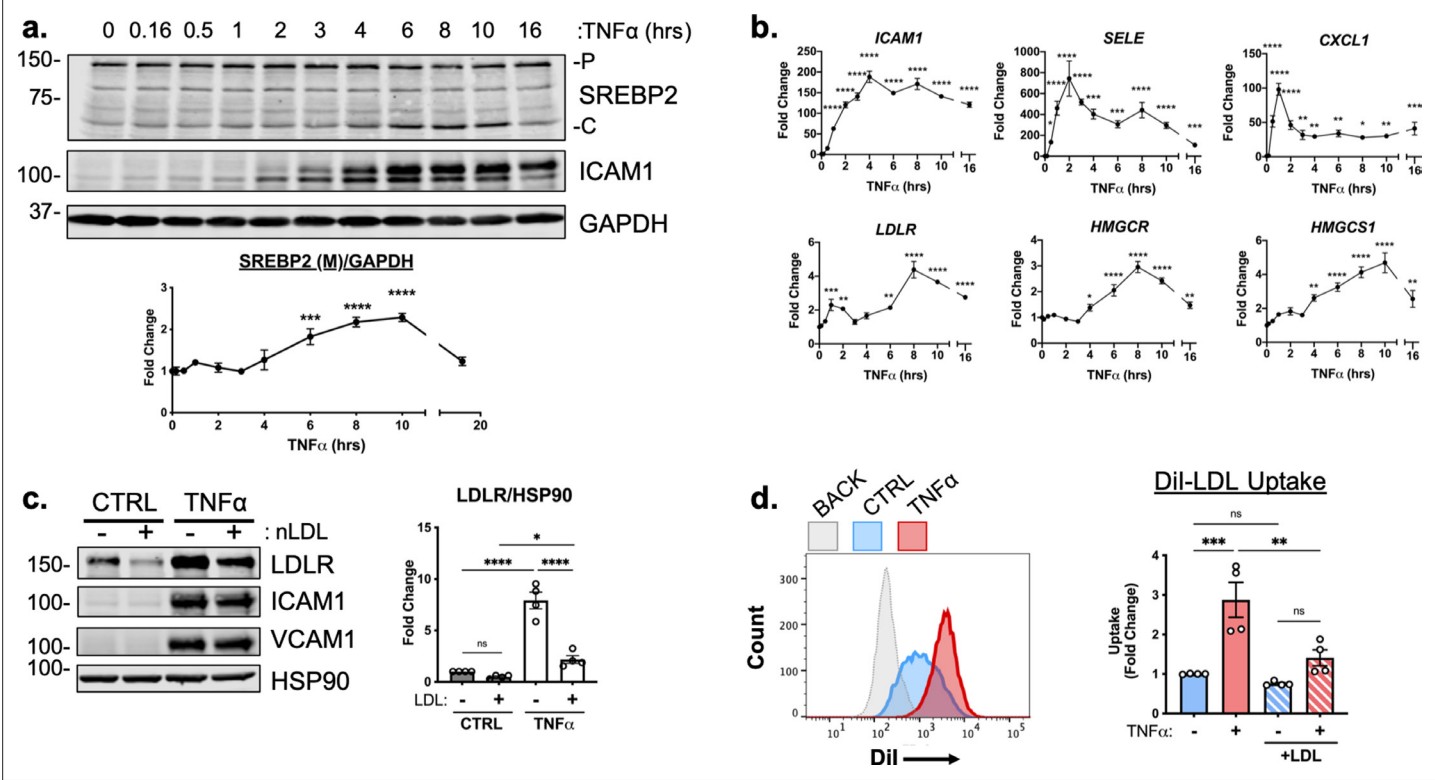

**Figure 2.** TNFα increases SREBP2 cleavage and transcription of canonical sterol-responsive genes. (**a**) SREBP2 immunoblot from whole-cell lysates from HUVEC treated with TNFα (10 ng/mL) for indicated time. Data are normalized to respective GAPDH and then to untreated cells (n=3). (**b**) qRT-PCR analysis of RNA from HUVEC treated with TNFα (10 ng/mL) for indicated time. Data are normalized to respective *ACTB* and then to untreated cells (n=8). (**c**) LDLR protein levels of TNFα-treated HUVEC treated with or without native LDL (25 µg/mL). Data are normalized to respective HSP90 levels and then to untreated cells (n=4). (**d**) Flow cytometry analysis of exogenous DiI-LDL uptake in HUVEC treated with TNFα and with indicated media. 2.5 µg/mL DiI-LDL was incubated for 1 hr at 37 °C before processing for flow cytometry. Uptake was quantified by PE mean fluorescence intensity per cell and normalized to untreated cells in LPDS across two experiments (10,000 events/replicate, n=4). *p<0.05; **p<0.01; ***p<0.001; ***p<0.0001 by one-way ANOVA with Dunnett's multiple comparisons test (**a and b**) or two-way ANOVA with Sidak's multiple comparisons test (**c and d**).

The online version of this article includes the following source data and figure supplement(s) for figure 2:

**Source data 1.** Blots corresponding to *Figure 2a and c*.

**Source data 2.** Raw data supporting *Figure 2a-d*.

**Figure supplement 1.** TNFa predominantly activates targets involved in cholesterol biosynthesis, not fatty acid synthesis.

**Figure supplement 1—source data 1.** Blots corresponding to *Figure 2—figure supplement 1a and c*.

and SREBP1 were significantly decreased in *RELA* knockdown cells. An analysis of gene set overlap between genes significantly upregulated after TNFα treatment and genes significantly downregulated in TNFα-treated cells lacking *RELA* revealed that SREBP2 target genes were significantly overrepresented (*Figure 1—figure supplement 1d*). We decided to focus on the SREBP2 pathway in late phase (10 hr) of TNFα-treated cells because of the overwhelming prevalence of cholesterol biosynthesis genes that were increased and significantly attenuated when *RELA* was knocked down (*Figure 1e*).

## TNFα increases SREBP2 cleavage and transcription of downstream gene targets

RNA-seq analysis predicted that SREBP2 was highly activated in HUVEC treated with TNFα. SREBP2 becomes transcriptionally active when its N-terminal DNA-binding fragment is proteolytically processed in the Golgi and allowed to enter the nucleus (*Sakai et al., 1996*). This process can be assayed by measuring precursor (P) and cleaved (C) SREBP2 at 150 kDa and 65 kDa on a Western Blot, respectively (*Hua et al., 1995*). Measurement of SREBP2(C) throughout a 16 hr timecourse revealed that SREBP2 cleavage began as early as 6 hr after TNFα treatment and peaked at 10 hr (*Figure 2a*).

Furthermore, SREBP2 activation was dose-dependently induced by TNFα in HUVEC cultured in either sterol-sufficient fetal bovine serum (FBS) or in lipoprotein depleted serum (LPDS) (*Figure 2—figure supplement 1a*).

We next measured the relative mRNA abundance of NF-κB and SREBP2 target genes throughout the same time course. Known EC NF-κB target genes, such as *ICAM1, SELE,* and *CXCL1* were quickly and robustly induced within 2 hr TNFα treatment, consistent with the notion that RNA polymerase II is primed at the promoters of these genes to elicit a rapid transcriptional response (*Figure 2b*, top) (*Adelman et al., 2009*). The patterning of SREBP2 target gene expression was notably several hrs later than NF-κB dependent gene expression. A majority of the canonical SREBP2 genes, including *LDLR, HMGCR, and HMGCS1* significantly increased as early as 6 hr after treatment and peaked at around 8–10 hr (*Figure 2b*, bottom). Unlike a majority of the cholesterol biosynthesis genes, several fatty acid synthesis genes known to be SREBP1-dependent, such as *ACACA/B, FASN*, and *GPAM* did not significantly increase with TNFα treatment and were unaffected by *RELA* knockdown (*Figure 2—figure supplement 1b*).

To further test if SREBP2 activity increased after TNFα stimulation, we measured the protein levels of low-density lipoprotein receptor (LDLR), a well-known target of SREBP2 target and receptor involved in the uptake of exogenous lipoproteins, such as low-density lipoprotein (LDL) (*Briggs et al., 1993*). HUVEC were treated overnight in LPDS with or without the addition of 25 μg/mL LDL, to suppress SREBP2 cleavage and LDLR expression (*Figure 2c*). LDL treatment decreased LDLR protein levels in HUVEC at rest and TNFα significantly increased LDLR levels in HUVEC cultured in both LPDS and LPDS +LDL. Exogenous LDL was also able to partially suppress LDLR expression, indicating that this process is sterol-sensitive. Similar results were found for another well-known SREBP2 target of cholesterol biosynthesis, HMGCR (*Figure 2—figure supplement 1c*). Next, we tested fluorescently labeled LDL uptake as a functional readout of the increase in LDLR as quantified by flow cytometry. Similar to what was seen by immunoblotting, TNFα treatment led to increased DiI-LDL uptake into cells pre-incubated in various degrees of sterol enriched media (*Figure 2d*).

## NF-κB activation and DNA binding are necessary for cytokine-induced SREBP2 cleavage

TNFα activates several signaling cascades to fully activate resting endothelial cells that leads to a change in phenotype to promote inflammation. For canonical NF-κB signaling, TNFα activates the immediate post-translational activation of the NF-κB complex via phosphorylation and degradation of the inhibitory molecule IκBα by I-κ-kinase (IKK) isoforms (*DiDonato et al., 1997*). However, TNFα has been shown to upregulate several other signaling pathways, such as JNK, p38, and ERK1/2 (*Aggarwal, 2003*). Therefore, we sought to confirm that NF-κB signaling is necessary for SREBP2 activation in ECs undergoing inflammatory stress. Treatment of HUVEC with TNFα, IL1β, and lipopolysaccharide (LPS) to activate NF-κB through separate pathways led to an increase in SREBP2 cleavage and upregulation of LDLR levels (*Figure 3a*) with concomitant increases in ICAM1 as a positive control. Furthermore, pre-treatment of HUVEC with the transcription inhibitor, Actinomycin D (ActD), blunted activation of SREBP2, LDLR and ICAM1 in response to TNFα (*Figure 3b, c*). This indicated that the mechanism by which inflammatory cytokines activate SREBP2 is most likely through the transcription of novel regulatory molecules rather than via post-translational modifications and/or processing.

Lastly, we measured TNFα-mediated SREBP2 activation after chemical inhibition and genetic knockdown of NF-κB to confirm previous RNA-seq results. Treatment of HUVEC with BAY 11–7082, a selective IKK inhibitor, significantly attenuated IL1β and TNFα induced increase in SREBP2 cleavage, LDLR protein levels, and mRNA expression of SREBP2-dependent genes (*Figure 3d and e*; *Keller et al., 2000*). Notably, BAY 11–7082 did not suppress JNK or p38 signaling, demonstrating specificity for the NF-κB pathway (*Figure 3d*). Western Blot analysis of SREBP2 in HUVEC after *RELA* knockdown confirmed that NF-κB DNA-binding and transcriptional activity are required for cytokine induction of SREBP2 cleavage (*Figure 3f*).

## Canonical SCAP/SREBP2 shuttling is required for TNFα-mediated SREBP2 cleavage

Studies have shown that SREBP2 cleavage can be controlled by mechanisms beyond the SCAP shuttling complex, such as Akt/mTOR/Lipin1 regulation of nuclear SREBP and direct cleavage of SREBP

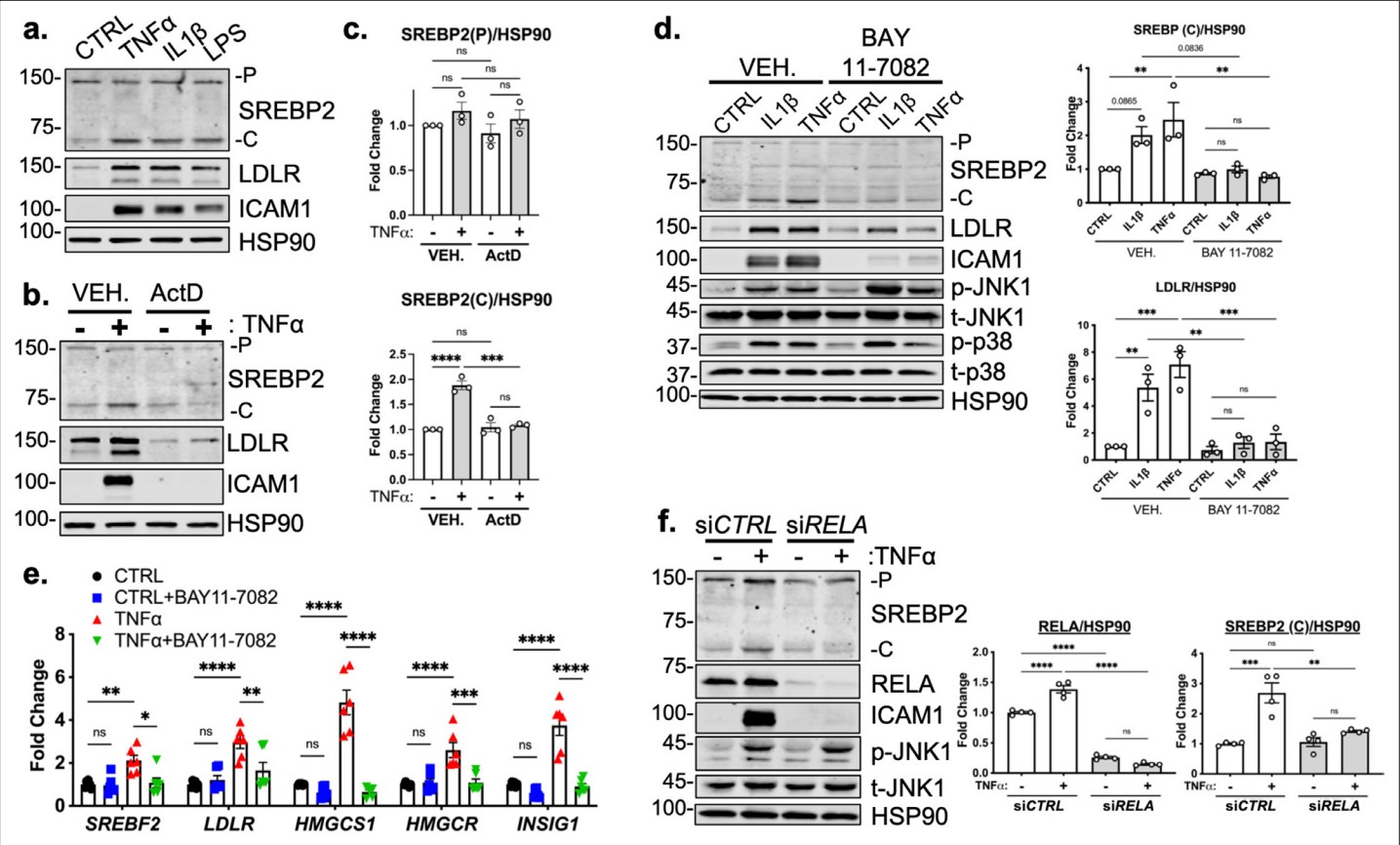

**Figure 3.** RELA DNA-binding is necessary for activation of SREBP2 by inflammatory stress. (**a**) Representative immunoblot of SREBP2 and LDLR protein levels in HUVEC treated with TNFα (10 ng/mL), IL1β (10 ng/mL), or LPS (100 ng/mL). (**b**) Representative immunoblot of SREBP2 and LDLR protein levels in HUVEC treated with actinomycin D (ActD, 10 ng/mL) and with or without TNFα (10 ng/mL). (**c**) Quantification of SREBP2 precursor (**p**) and cleaved (**c**) from (**b**). Data are normalized to respective HSP90 and then to untreated cells (n=4). (**d**) SREBP2 and LDLR protein levels in HUVEC treated with IL1β (10 ng/mL) or TNFα (10 ng/mL) and with or without NF-κB inhibitor, BAY11-7082 (5 µM). Data are normalized to respective HSP90 and then to untreated cells (n=3). (**e**) qRT-PCR analysis of SREBP2-dependent genes, *SREBF2, LDLR, HMGCS1, HMGCR,* and *INSIG1,* expression in HUVEC treated with or without TNFα (10 ng/mL) and BAY11-7082 (5 µM). Data are normalized to respective *ACTB* and then to untreated cells (n=6). (**f**) SREBP2 and RELA levels in TNFα (10 ng/mL)-treated HUVEC treated with or without siRNA targeting *RELA*. Data are normalized to respective HSP90 and then to untreated cells (n=4). *p<0.05; **p<0.01; ***p<0.001; ***P<0.0001 by one-way ANOVA (c, d, and f) or two-way ANOVA (**e**) with Tukey's multiple comparison's test.

The online version of this article includes the following source data for figure 3:

**Source data 1.** Blots corresponding to *Figure 3a, b, d and f*.

**Source data 2.** Raw data supporting *Figure 3c, d, e and f*.

in the ER by S1P (*Shimano and Sato, 2017*; *Kim et al., 2018*). It is possible that a post-translational SREBP2 regulator could be the RELA-dependent molecule responsible for increased SREBP2 activation. Furthermore, it is also feasible that the *SREBF2* gene itself is under the control of NF-κB, which would upregulate total SREBP2 and increase the threshold of cholesterol needed to suppress its cleavage. Interestingly, this has been reported in a previous study for *SREBF1* (*Im et al., 2011*). Thus, we used several SREBP-processing inhibitors to test if the SCAP-mediated translocation and Golgi cleavage are necessary for SREBP2 activation in cytokine stimulated EC (*Figure 4a*).

Upon sensing heightened cellular cholesterol, SCAP stabilizes SREBP in the ER and prevents its translocation to the Golgi for processing (*Brown and Goldstein, 1997*). Therefore, we treated HUVEC with two forms of exogenous cholesterol to test if SCAP shuttling lies upstream of SREBP2 activation in our system: [1] free cholesterol bound to the donor molecule methyl-β-cyclodextrin (Chol) and [2] cholesterol-rich LDL. Chol significantly attenuated the increased LDLR and SREBP2 cleavage seen with TNFα stimulation (*Figure 4b*). Furthermore, LDL was able to dose-dependently decrease SREBP2 activation back down to basal levels at the highest concentration of 250 µg/mL (*Figure 4c*,

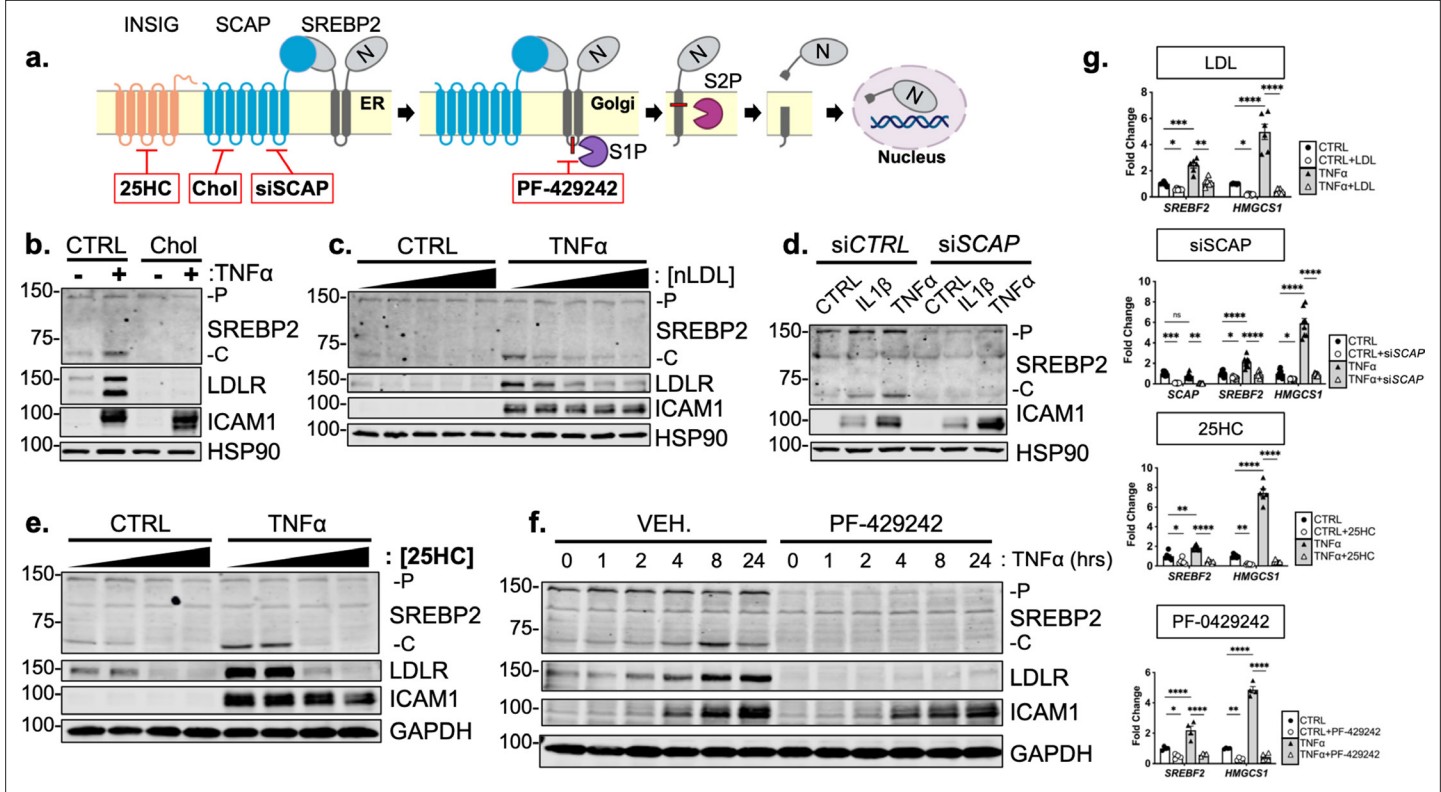

**Figure 4.** Cytokine-mediated upregulation of SREBP2 cleavage requires proper SCAP shuttling and proteolytic processing in the Golgi. (**a**) Schematic of where 25-hyroxycholesterol (25HC), cholesterol, siSCAP, and PF-429242 inhibit SREBP processing throughout the pathway. (**b**) Representative immunoblot of SREBP2 and LDLR protein levels in HUVEC treated with TNFα (10 ng/mL) and cholesterol (Chol) (25 μg/mL). Data are normalized to respective HSP90 and then to untreated cells. (**c**) Representative immunoblot of SREBP2 and LDLR protein levels in HUVEC treated with TNFα (10 ng/mL) and increasing concentrations of LDL. Data are normalized to respective HSP90 and then to untreated cells. (**d**) Representative immunoblot SREBP2 cleavage in HUVEC treated with IL1β (10 ng/mL) or TNFα (10 ng/mL) and *SCAP* siRNA. Data are normalized to respective HSP90 and then to untreated cells. (**e**) Representative immunoblot of SREBP2 and LDLR protein levels in HUVEC treated with TNFα (10 ng/mL) and increasing concentrations of 25-hydroxycholesterol (25HC). Data are normalized to respective HSP90 and then to untreated cells. (**f**) Representative immunoblot of SREBP2 and LDLR protein levels in HUVEC treated with TNFα (10 ng/mL) and PF-429242 (10 μM) for indicated time. Data are normalized to respective HSP90 and then to untreated cells. (**g**) qRT-PCR analysis of *SREBF2, HMGCS1,* and *SCAP* from RNA of HUVECs treated with TNFα (10 ng/mL) and indicated SREBP2 inhibitor. Data are normalized to respective *ACTB* and then to untreated cells (n=6). *p<0.05; **p<0.01; ***p<0.001; ***p<0.0001 by two-way ANOVA with Sidak's multiple comparisons test.

The online version of this article includes the following source data and figure supplement(s) for figure 4:

**Source data 1.** Blots corresponding to *Figure 4b, c, d, e and f*.

**Source data 2.** Raw data supporting *Figure 4g* and *Figure 4—figure supplement 1 a, c, d, e, and f*.

**Figure supplement 1.** Immunoblots of SREBP2 processing inhibitors at effective doses.

**Figure supplement 1—source data 1.** Blots corresponding to *Figure 4—figure supplement 1a, b, c, and d*.

*Figure 4—figure supplement 1a*). Similar results were seen when SCAP was inhibited with a chemical inhibitor, fatostatin (*Figure 4—figure supplement 1b*). To solidify this point, siRNA knockdown of *SCAP* inhibited the ability of IL1β and TNFα to upregulate SREBP2 cleavage, but not ICAM1 induction (*Figure 4d*).

Next, we sought to inhibit SREBP2 processing by two complimentary approaches, INSIG1-mediated retention in the ER and inhibition of Golgi processing. The oxysterol 25-hydroxycholesterol (25HC) will promote association of INSIG to the SCAP/SREBP2 complex and prevent translocation to the Golgi (*Radhakrishnan et al., 2007*). Treatment of HUVEC with 25HC significantly prevented cytokine-induced SREBP2 activation and LDLR upregulation (*Figure 4e*, *Figure 4—figure supplement 1c*). We next treated the cells with PF-429242, a potent inhibitor of site-1-protease (S1P), which prevented SREBP2 cleavage and LDLR increase throughout the 24 hr timecourse (*Figure 4f*, *Figure 4—figure supplement 1 d*). The above biochemical experiments were supported by qPCR measurements of

SREBP2-dependent genes to confirm that the inhibitors used in this study fully attenuated SREBP2 activity (*Figure 4g*). As expected, *HMGCS1* mRNA was depleted basally by LDL, siSCAP, 25HC, and PF-042424 and these compounds prevented the increase in *HMGCS1* transcription in response to TNFα. *HMGCS1* transcript levels represented the trend seen in several other sterol-responsive genes. Although TNFα consistently increased *SREBF2* transcription, all inhibitors also were able to attenuate this upregulation of *SREBF2* mRNA. Taken together, the evidence suggests that canonical SCAP shuttling is necessary for activation of SREBP2 by inflammatory cytokines and that this is not due to direct NF-κB-mediated upregulation of the *SREBF2* transcript or cholesterol biosynthesis genes.

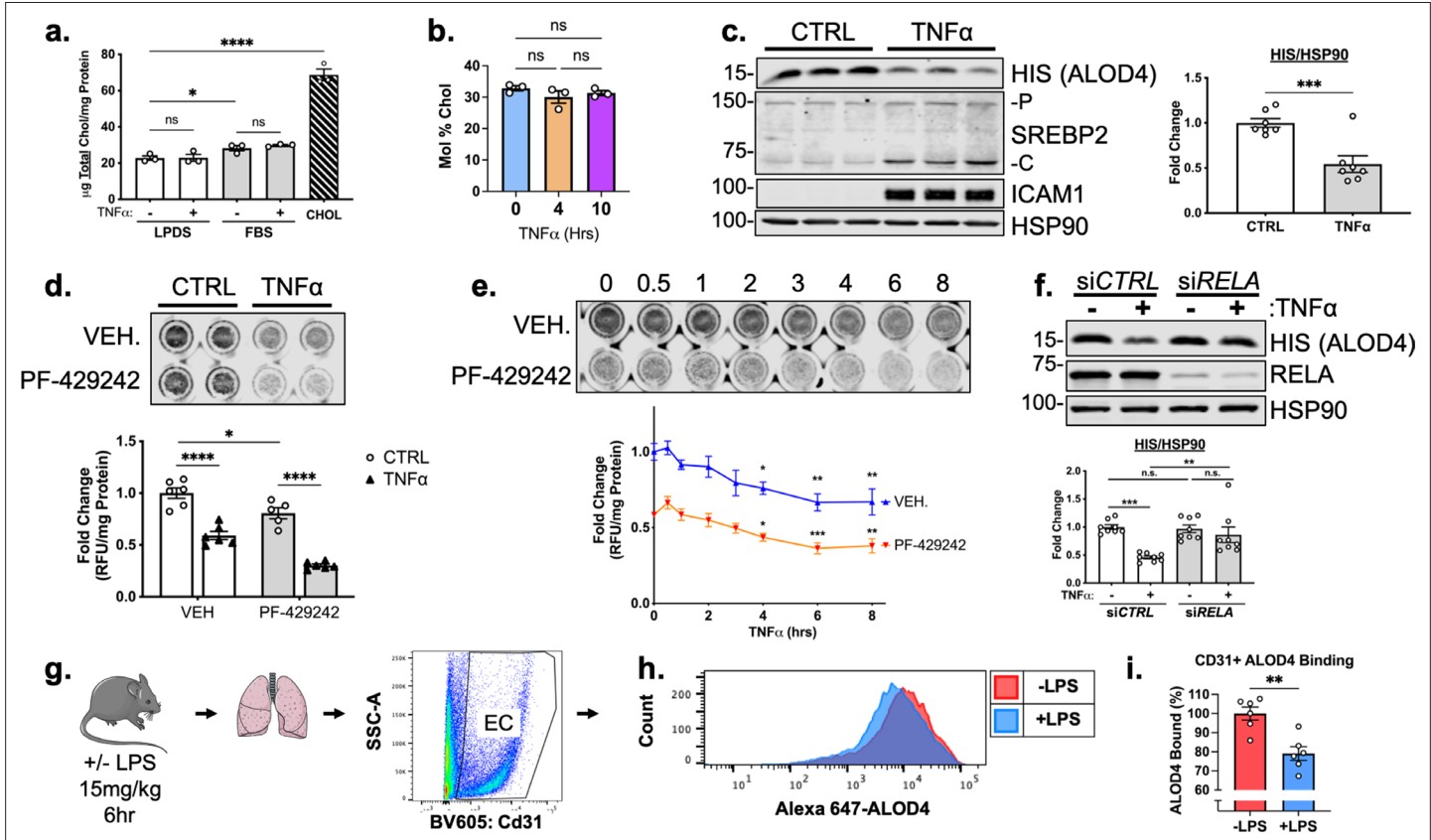

**Figure 5.** TNFα decreases accessible cholesterol in cultured HUVEC and mouse lung ECs in vivo. (**a**) Quantification of total cholesterol extracted from HUVEC treated with or without TNFα (10 ng/mL) and indicated positive controls, lipoprotein deficient serum (LPDS), fetal bovine serum (FBS), or MβCD-cholesterol. Data were normalized to respective total protein (n=3). (**b**) Total cholesterol in HUVEC after 4 or 10 hr of TNFα (10 ng/mL) quantified by mass spectrometry (n=3). (**c**) ALOD4 protein levels in HUVEC treated with TNFα (10 ng/mL). Data are normalized to respective HSP90 and then to untreated cells (n=7). (**d**) In-cell western blot of ALOD4 protein levels in HUVEC treated with TNFα (10 ng/mL) and PF-429242 (10 µM). Data are normalized to respective total protein and then to untreated cells (n=6). (**e**) In-cell western blot of ALOD4 protein levels in HUVEC treated with TNFα (10 ng/mL) and PF-429242 (10 µM) for indicated time. Data are normalized to respective total protein and then to untreated cells (n=6). (**f**) ALOD4 protein levels in TNFα (10 ng/mL)-treated HUVEC treated with or without *RELA* siRNA. Data are normalized to respective HSP90 and then to untreated cells (n=8). (**g**) Schematic of protocol to isolate mouse lung endothelial cells and quantify ALOD4 binding by flow cytometry. (**h**) Representative histogram of ALOD4 binding in Cd31 +lung endothelial cells in mice treated with or without LPS (15 mg/kg) for 6 hr. (**i**) Quantification of ALOD4 binding across 2 flow cytometry experiments in mice treated with or without LPS (15 mg/kg). Binding was quantified as AlexaFluor647 mean fluorescent intensity per cell (100,000 events/replicate). Data are normalized to nontreated mice (-LPS, n=6;+LPS, n=6). *p<0.05; **p<0.01; ***p<0.001; ***p<0.0001 by one-way ANOVA with Tukey's multiple comparison's test (**a and d**) or Dunnett's multiple comparisons test (**e**), unpaired t-test (**c and i**), or two-way ANOVA with Sidak's multiple comparisons test (**f**).

The online version of this article includes the following source data and figure supplement(s) for figure 5:

**Source data 1.** Blots corresponding to *Figure 5c and f*.

**Source data 2.** Raw data supporting *Figure 5a, c, d, e, f and i* and *Figure 5—figure supplement 1e, f, g, and h*.

**Figure supplement 1.** Optimization of assays used to quantify accessible cholesterol on cellular plasma membranes.

**Figure supplement 1—source data 1.** Blots corresponding to *Figure 5—figure supplement 1a and c*.

## Inflammatory stress decreases accessible free cholesterol required for SREBP processing

Since SCAP/SREBP2 shutting is maintained when cells were treated with TNFα, we reasoned that perhaps TNFα regulates cellular cholesterol levels. Lipids were extracted from control and TNFα-treated HUVEC and total cholesterol was measured. Incubation of cells in LPDS reduced cellular cholesterol compared to cells cultured in FBS. Cells treated with exogenous methyl-β-cyclodextrin-cholesterol (Chol) contained significantly more measured cholesterol (*Figure 5a*). However, TNFα treatment did not alter total cholesterol in either media condition. Secondly, we quantified cellular cholesterol using mass spectrometry-based lipid analysis, a significantly more precise technique that provides information on molar percentages of lipid. Similar to the initial cholesterol measurements, TNFα did not change total cholesterol after 4 or 10 hr of treatment (*Figure 5b*).

Changes in the distribution of cholesterol could account for SREBP2 activation without loss in total cholesterol mass. Recently, several tools have been developed to analyze the exchangeable pool of free cholesterol that exists in flux between the ER and the plasma membrane and this accessible pool tightly regulates the shuttling of SCAP/SREBP (*Infante and Radhakrishnan, 2017*). Accessible cholesterol can be quantified using modified recombinant bacterial toxins that bind in a 1:1 molar ratio to accessible free cholesterol on the plasma membrane (*Gay et al., 2015*). To examine accessible cholesterol in EC, we purified His-tagged anthrolysin O (ALOD4) and used it as a probe (*Endapally et al., 2019a*). To assess the utility of the probe in HUVEC, cells were incubated with Chol, LDL or with methyl-β-cyclodextrin (MβCD, a cholesterol acceptor) and ALOD4 bound was quantified by probing for anti-HIS at 15 kDa by western blotting. As expected, treatment with Chol or LDL increased ALOD4 binding, whereas MβCD decreased ALOD4 binding (*Figure 5—figure supplement 1a*). Secondly, a similar method was used, but instead of cell lysis, DyLight680-conjugated anti-HIS antibody was directly applied to the ALOD4-incubated cells and read live on LICOR Biosciences Odyssey CLx platform (In-Cell Western Blot). Likewise, the positive controls were able to tightly regulate ALOD4 binding and fluorescence signal (*Figure 5—figure supplement 1b*).

TNFα treatment of HUVEC significantly decreased ALOD4 binding (*Figure 5c*). Using In-Cell Western blotting, treatment with PF-429242 to reduce SREBP2 processing and its transcription decreased ALOD4 basally and, when combined with TNFα, significantly decreased accessible cholesterol even further (*Figure 5d*). This suggests that the decrease in accessible cholesterol was independent of SREBP2 stability. Probing accessible cholesterol throughout an 8 hr timecourse revealed that ALOD4 binding significantly decreased as early as 4 hr after TNFα treatment in HUVEC treated with and without PF-429242 (*Figure 5e*). This was in line with previous results measuring SREBP2 cleavage and gene expression as accessible cholesterol depletion should precede SREBP2 activation. Moreover, *RELA* knockdown attenuated the decrease in accessible cholesterol, indicating that SREBP2 activation by NF-κB was most likely through upregulation of a molecule or pathway that decreases accessible cholesterol (*Figure 5f*).

Lastly, we validated that inflammatory stress decreased accessible cholesterol not only in cultured HUVEC, but also in EC in vivo. We directly labeled ALOD4 with AlexaFluor 647 and incubated this probe with suspended HUVEC to validate the flow cytometry assay (*Figure 5—figure supplement 1c and d*). Modulation of cholesterol with various treatments altered ALOD4 binding as expected and treatment of HUVEC with TNFα revealed similar results to immunoblotting (*Figure 4—figure supplement 1e and f*). Next, we intraperitoneally injected wildtype C57BL/6 J mice with a nonlethal dose of LPS at 15 mg/kg to stimulate a systemic inflammatory response. Indeed, TNFα peaked in the serum of these animals 2 hr after injection (*Figure 5—figure supplement 1g*). Lungs were harvested 6 hr after LPS injection and cells were broken up into a single-cell suspension for flow cytometry staining (*Figure 5g*). Cd31 +ECs from mice treated with LPS contained about 20% less accessible cholesterol compared to ECs from untreated mice (*Figure 5h and i*). Notably, total serum cholesterol remained unchanged in TNFα-treated animals compared to control, indicating that the decrease in accessible cholesterol reflects the effect of the inflammatory stimulus on ECs (*Figure 5—figure supplement 1h*).

## STARD10 is necessary for full SREBP2 activation by TNFα

TNFα did not impact several canonical biological pathways that regulate the pool of accessible cholesterol. Briefly, cholesterol efflux, sphingomyelin shielding, esterification, and lysosomal/endosomal accumulation remained unchanged in ECs treated with TNFα (*Figure 6—figure supplement*

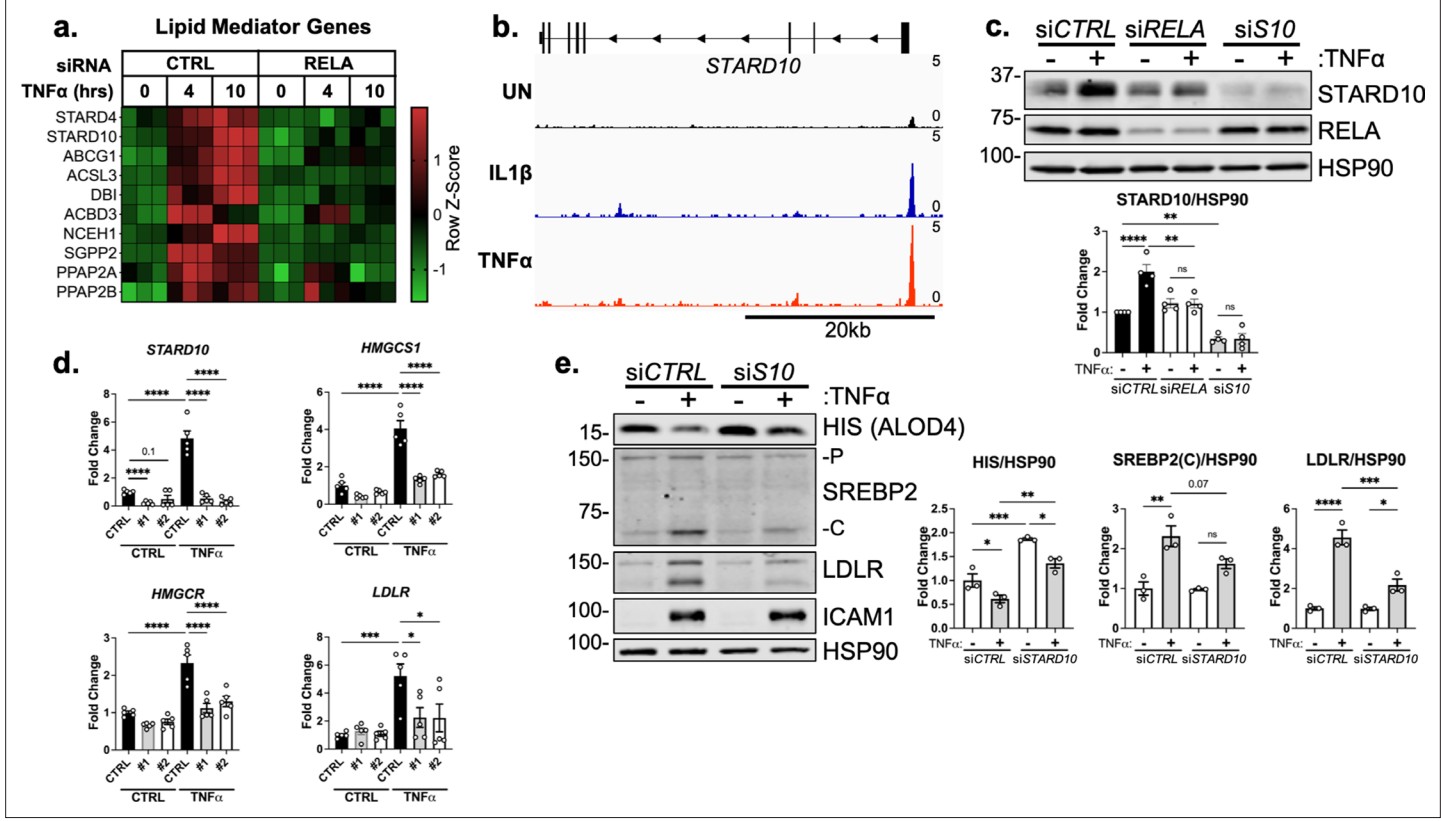

**Figure 6.** STARD10 is necessary for complete TNFα-mediated accessible cholesterol reduction and SREBP2 activation. (**a**) Heatmap of genes that regulate lipid homeostasis, significantly increased with TNFα (10 ng/mL) treatment after 4 or 10 hr, and were significantly inhibited by *RELA* knockdown. (**b**) *STARD10* gene locus from RELA ChIP-seq analysis of human aortic endothelial cells (HAEC) treated with TNFα (2 ng/mL) or IL1β (10 ng/mL) for 4 hr. Data are scaled from 0 (bottom) to 5 (top). Data originated from GSE89970. (**c**) Immunoblot of STARD10 protein levels in HUVEC treated with *RELA* or *STARD10* (**S10**) siRNA and with or without TNFα (10 ng/mL). Data are normalized to respective HSP90 levels and then to untreated cells (n=4). (**d**) qRT-PCR analysis of RNA from HUVEC treated with TNFα (10 ng/mL) and two independent siRNA targeting *STARD10* (#1,#2). Data are normalized to respective *ACTB* and then to untreated cells (n=5). (**e**) Immunoblot of ALOD4, SREBP2, and LDLR protein levels in HUVEC treated with *STARD10* siRNA (si*S10*) and with or without TNFα (10 ng/mL). Data are normalized to respective HSP90 levels and then to untreated cells (n=3). *p<0.05; **p<0.01; ***p<0.001; ***p<0.0001 by two-way ANOVA with Sidak's multiple comparisons test (**d** and **e**).

The online version of this article includes the following source data and figure supplement(s) for figure 6:

**Source data 1.** Blots corresponding to *Figure 6c and e*.

**Source data 2.** Raw data supporting *Figure 6c, d and e*, *Figure 6—figure supplement 1* c and e, and *Figure 6—figure supplement 3c*.

**Figure supplement 1.** Complete analysis of common cholesterol transport mechanisms in HUVEC under inflammatory stress.

**Figure supplement 1—source data 1.** Blots corresponding to *Figure 6—figure supplement 1c, d, and f*.

**Figure supplement 2.** RELA ChIP-seq analysis of lipid mediator genes.

**Figure supplement 3.** ABCG1 is significantly upregulated by TNFa, but is not responsible for accessible cholesterol depletion or SREBP2 activation.

**Figure supplement 3—source data 1.** Blots corresponding to *Figure 6—figure supplement 3a and b*.

1). Therefore, we probed our RNA-seq dataset for genes that have been reported to regulate lipid dynamics and could possibly be upstream of the depletion in accessible cholesterol.

We specifically isolated genes that significantly increased after 4 or 10 hr TNFα treatment and decreased with *RELA* knockdown. We found several genes that perform various lipid-associated functions, such as direct lipid binding and transport (*STARD4, STARD10,* and *ABCG1*), free fatty acid enzymatic activation and transport (*ACSL3, DBI*), mediation of mitochondrial steroidogenesis (*DBI, ACBD3, NCEH1*), and metabolism of phospholipids (*SGPP2, PAPP2A,* and *PPAPP2B*) (*Figure 6a*). We analyzed previously submitted RELA chromatin immunoprecipitation (ChIP) sequencing data of ECs treated with IL1β or TNFα to identify gene candidates that were direct targets of NF-κB (*Hogan et al., 2017*). Of these genes, *ABCG1, STARD10, DBI, and NCEH1* appeared to have increased RELA occupancy

within their promoters upon inflammatory stimulation (*Figure 6—figure supplement 2*). Although our results indicate that inflammatory stress did not increase cholesterol efflux, we were particularly interested in *ABCG1* because of its known role in cholesterol homeostasis and transport (*Kennedy et al., 2005*; *Tarling and Edwards, 2011*). Indeed, ABCG1 protein levels drastically increased in ECs treated with TNFα, which was attenuated by chemical inhibition of NF-κB (*Figure 6—figure supplement 3a*). ABCA1, a close family member to ABCG1 that shares LXR as its key transcription regulator, did not increase with inflammatory cytokine (*Venkateswaran et al., 2000*). However, knockdown of *ABCG1* by two independent siRNA were unable to rescue the effect of TNFα on accessible cholesterol or SREBP2 activation (*Figure 6—figure supplement 3b and c*). Nonetheless, the robust activation of ABCG1 by inflammatory cytokines warrants further study in the EC inflammatory response.

From these several targets identified by RNA-seq and ChIP-seq, we identified *STARD10* as a promising upstream mediator of accessible cholesterol in ECs treated with inflammatory cytokines. STARD10 belongs to a family of proteins that bind hydrophobic lipids via a structurally conserved steroidogenic acute regulatory-related lipid transfer (START) domain (Clark, 2020). STARD proteins regulate nonvesicular trafficking of cholesterol, phospholipids, and sphingolipids between membranes. *STARD10* was found to be upregulated by TNFα after 4 and 10 hr of treatment and inhibited by loss of *RELA* (*Figure 6a*). *STARD4* also shared this expression pattern, however, it is a sterol-sensitive gene and its upregulation by TNFα most likely occurred via SREBP2 activation (*Soccio et al., 2005*). Analysis of RELA chromatin ChIP-seq data revealed increased occupancy of RELA within the *STARD10* promoter in ECs treated with IL1β or TNFα (*Figure 6b*). Immunoblotting confirmed what was indicated by the RNA-seq and ChIP-seq results, revealing that STARD10 was significantly upregulated by TNFα and its expression was dependent on RELA (*Figure 6c*). STARD10 has been shown to bind phosphatidylcholine (PC), phosphatidylethanolamine (PE), and phosphatidylinositol (PI), but much of its detailed biology remains unknown (*Olayioye et al., 2005*; *Carrat et al., 2020*). Although it has not been shown to directly bind cholesterol, STARD10 may be implicated in the reorganization of membrane phospholipids that could alter cholesterol flux and shield cholesterol localization (*Tabas, 2002*; *Mesmin and Maxfield, 2009*; *Lagace, 2015*).

We next knocked down STARD10 to analyze its role in cholesterol homeostasis and EC inflammatory response. Treatment of ECs with two independent siRNAs targeting STARD10 significantly decreased *STARD10* expression and attenuated the enhanced expression of SREBP2 target genes *HMGCS1, HMGCR,* and *LDLR* (*Figure 6d*). Furthermore, STARD10 knockdown significantly rescued the loss in accessible cholesterol that occurs with TNFα stimulation (*Figure 6e*). SREBP2 activation and LDLR upregulation were also attenuated with STARD10 silencing. Therefore, we have identified a novel RELA-inducible gene in ECs that mediates the cholesterol homeostasis in response to inflammatory stress.

## Discussion

Little is known about cholesterol metabolism and homeostasis in ECs in the context of physiology or pathology. Here, we show that TNFαand IL-1β induce a transcriptional response via NF-κB that alters cholesterol homeostasis in ECs (*Figure 7*). Accordingly, rapid changes in accessible cholesterol then activates SREBP2 processing to compensate for the altered flux of accessible cholesterol between the plasma membrane and ER. Importantly, we identified a novel NF-κB-inducible gene, *STARD10*, that serves as an intermediate bridging TNF activation to changes in accessible cholesterol and SREBP2 activation. Collectively, these data support the growing evidence for an intimate relationship between inflammatory signaling and cholesterol homeostasis.

Different forms of cellular stress have been shown to activate SREBP2 in multiple cell types, including oscillatory shear stress in ECs, ER stress in hepatocytes, and TNFα in macrophages (*Xiao et al., 2013*; *Kim et al., 2018*; *Kusnadi et al., 2019*). However, these studies did not implicate changes in cholesterol accessibility as an upstream mechanism leading to the activation of SREBP2. Here, we clearly demonstrate that activation of NF-kB leads to changes in the accessible cholesterol pool promoting classical SREBP2 processing in a SCAP-dependent manner. Epithelial cells treated with IFNγ-stimulated macrophage conditioned media rapidly reduce accessible cholesterol to protect against bacterial infection by preventing cholesterol-mediated transport between cells (*Abrams et al., 2020*). Similarly, macrophages deplete accessible cholesterol to protect against bacterial toxin-mediated injury (*Zhou et al., 2020*). In both of these studies, accessible cholesterol was mediated by

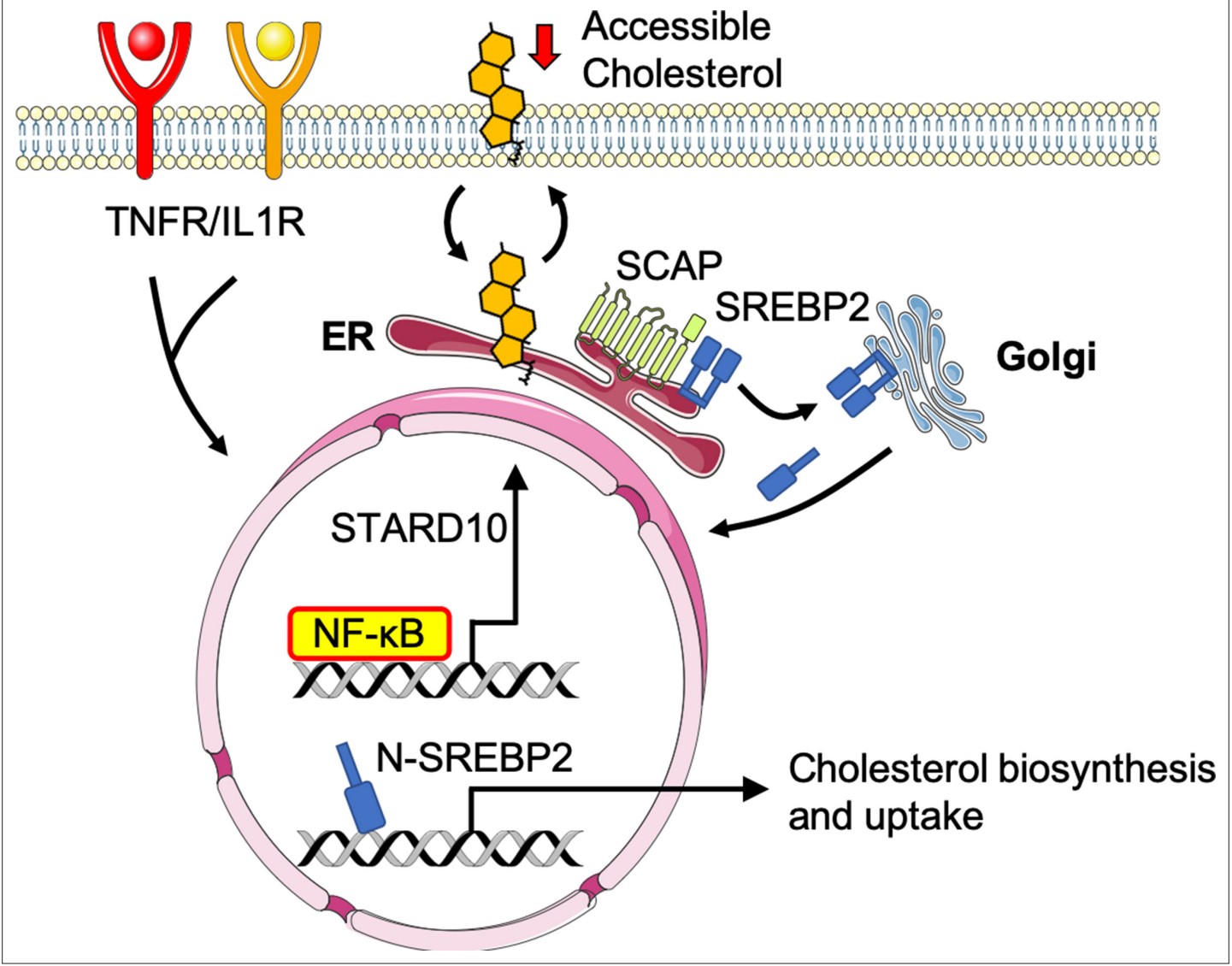

**Figure 7.** Working model of the relationship between sterol sensing and EC acute inflammatory response. Pro-inflammatory cytokines, such as TNFα and IL1β, promote NF-κB activation of gene transcription in endothelial cells. NF-κB upregulates factors, such as *STARD10*, that significantly decrease accessible cholesterol on the plasma membrane. SCAP senses the reduction in accessible cholesterol and shuttles SREBP2 to the Golgi to initiate classical proteolytic processing. Active N-SREBP2 translocates to the nucleus to transcriptionally upregulate canonical cholesterol biosynthetic genes.

IFN-induction of the enzyme, cholesterol 25-hydroxylase (Ch25H), which produces 25-HC as a product and enhances the esterification of cholesterol. However, Ch25H is not expressed ECs with or without TNFα and cholesterol does not mobilize into the lipid droplet pool. Nonetheless, EC depletion of accessible cholesterol may be an evolutionary mechanism of host immunity.

In the present study, we show that TNFα activation of NF-κB induces the expression of the gene, STARD10. Although little is known about the physiological function of STARD10, it may influence cholesterol homeostasis in several ways. Firstly, it has been reported that STARD10 can directly bind PC, PE, and PI and may alter intracellular membrane dynamics to influence the flux of cholesterol. It has been suggested PC plays a role in cholesterol sequestration either by steric hindrance from by its polar head group or through the creation of novel membranes that may sink cholesterol out of the accessible pool (*Mesmin and Maxfield, 2009*; *Lagace, 2015*). Furthermore, STARD10 belongs to a classical family of lipid transporters and may bind lipids not previously reported. As a novel NFκB inducible gene, STARD10 regulation of cholesterol homeostasis is a novel concept that warrants further investigation.

Previous studies indicate that SREBP2 activation feeds forward into the inflammatory response and exacerbates inflammatory damage. Firstly, SREBP2 has been shown to regulate inflammatory phenotype via modulation of cholesterol homeostasis. Increased cholesterol flux has been reported to feed into multiple immune pathways, such as interferon responses, inflammasome activation, and trained immunity (*York et al., 2015*; *Dang et al., 2017*; *Bekkering et al., 2018*). Furthermore, perturbations in cellular cholesterol may change membrane dynamics and affect cellular signaling (*Araldi et al., 2017*). Secondly, SREBP2 has been proposed to bind and promote transcription of several pro-inflammatory mediators, such as *IL1β, IL8, NLRP3,* and *NOX2* (*Kusnadi et al., 2019*; *Xiao et al., 2013*; *Yeh et al., 2004*). SREBP2 binding to non-classical gene promotors may very well depend on cellular and epigenetic context. How SREBP2 feeds into post-translational inflammatory response phenotype of ECs, such as permeability, sensitivity to infection, and receptor signaling, remains to be explored.

This study extends a growing body of work identifying a tight relationship between cholesterol homeostasis, inflammation and immunity. Although we have shown EC response to inflammatory cytokines in the acute setting, much remains to be explored in the context of chronic inflammation. Of particular interest, it has been well appreciated that the endothelium plays an important role in the progression of atherosclerosis. Chronic exposure to elevated lipoproteins causes accumulation of LDL in the subendothelial layer and activation of the endothelium (*Libby et al., 2019*). This leads to an inflammatory cascade that causes a cycle of leukocyte recruitment and inflammatory activation. In this pathological context, ECs are exposed to a unique microenvironment composed of relatively high concentrations of cholesterol and cytokines. Thus, elucidating how the endothelium in vivo responds to the loss of *Srebf2* in mouse models of acute and chronic inflammation will be important delineate the role of EC cholesterol homeostasis in vascular health and disease.

## Materials and methods

### Mammalian cell culture

HUVECs were obtained from the Yale School of Medicine, Vascular Biology and Therapeutics Core facility. Cells were cultured in EGM-2 media (Lonza) with 10% fetal bovine serum (FBS), penicillin/streptomycin and glutamine (2.8 mM) in a 37 °C incubator with 5% $CO_2$ supply.

### RNA sequencing

RNA was isolated using the RNeasy Plus Kit (Qiagen) and purity of total RNA per sample was verified using the Agilent Bioanalyzer (Agilent Technologies, Santa Clara, CA). RNA sequencing was performed through the Yale Center for Genome Analysis using an Illumina HiSeq 2000 platform (paired-end 150 bp read length). Briefly, rRNA was depleted from RNA using Ribo-Zero rRNA Removal Kit (Illumina). RNA libraries were generated from control cells using TrueSeq Small RNA Library preparation (Illumina) and sequenced for 45 cycles on Illumina HiSeq 2000 platform (paired end, 150 bp read length).

### RNA-seq analysis

Normalized counts and gene set enrichment analysis statistics were generated with Partek Flow. Reads were aligned to the hg19 build of the human genome with STAR and quantified to an hg19 RefSeq annotation model through Partek E/M. Gene counts were normalized as counts per million (CPM) and differential analysis was performed with GSA. Ingenuity Pathway Analysis (Ingenuity Systems QIAGEN) software was used to perform Canonical Pathway and Upstream Regulator analyses (Cutoff: p<0.05; –1.5>Fold Change >1.5). Metabolic network analysis was done using MetaCore (Clarivate) (Cutoff: p<0.005). GSEA analysis was used to produce Hallmark gene sets (1000 permutations, collapse to gene symbols, permutate to phenotype). Data are deposited in NCBI Gene Expression Omnibus and are available under GEO accession GSE201466.

### Western blotting analysis

Cells or tissues were lysed on ice with ice-cold lysis buffer containing 50 mM Tris-HCl, pH 7.4, 0.1 mM EDTA, 0.1 mM EGTA, 1% Nonidet P-40, 0.1% sodium deoxycholate, 0.1% SDS, 100 mM NaCl, 10 mM NaF, 1 mM sodium pyrophosphate, 1 mM sodium orthovanadate, 1 mM Pefabloc SC, and 2 mg/ml protease inhibitor mixture (Roche Diagnostics) and samples prepared. Total protein (25 µg) was loaded

into SDS-PAGE followed by transfer to nitrocellulose membranes. Immunoblotting was performed at 4 °C overnight followed by 1 hr incubation with LI-COR compatible fluorescent-labeled secondary antibodies (LI-COR Biosciences). Bands were visualized on the Odyssey CLx platform (LICOR Biosciences). Quantifications were based on densitometry using ImageJ.

## Quantitative RT-qPCR

RNA from cells or tissues were isolated using the RNeasy Plus Kit (Qiagen). 0.5 mg RNA/sample was retrotranscribed with the iScript cDNA Synthesis Kit (BioRad). Real-time quantitative PCR (qPCR) reactions were performed in duplicate using the CFX-96 Real Time PCR system (Bio-Rad). Quantitative PCR primers were designed using Primer3 software and synthesized by Yale School of Medicine Oligo Synthesis facility. Fold changes were calculated using the comparative Ct method.

## DiI-LDL uptake

Cells were washed in PBS and treated for 1 hr with plain EBM-2 containing 2.5 µg/mL DiI-LDL (Kalen Biomedical). Cells were washed for 5 min with acid wash (25 mM Glycine, 3% (m/V) BSA in PBS at pH 4.0), before suspended in PBS, washed, and fixed. PE mean fluorescence intensity per cell was measured by LSRII (BD Biosciences) flow cytometer the same day of the assay and analyzed using FlowJo.

## Thin layer chromatography (TLC)

Dried lipids were resuspended in hexane and loaded onto a silica gel TLC 60 plate (Millipore Sigma) and run in hexane:diethyl ether:acetic acid (70:30:1) until the solvent line reached approximately 1 inch from the top. Standards of pure triglycerides, diacylglycerides, cholesterol, and cholesterol ester were loaded for reference. After drying, the plate was exposed to a phosphor screen for 1 week and imaged using a Typhoon phosphorimager.

## Cholesterol efflux assay

Cells were equilibrated with 1 µCi/mL 3H-cholesterol (PerkinElmer) for 16 hr in full media containing FBS and ACAT inhibitor 58035 (Sigma). Next, cells were washed twice with PBS and incubated for 6 hr in serum-free media containing 58035 and indicated cholesterol acceptor. Media and cell lysis were harvested at the end of 6 hr. Ultima Gold scintillation liquid (PerkinElmer) were added to the media and cell lysis, respectively, and radioactivity was quantified using a Tri-Carb 2100 liquid scintillation counter (PerkinElmer). Efflux was measured as percent counts in media divided by counts in the cell lysis.

## Filipin staining

Confluent HUVEC cells were fixed and stained with 50 µg/mL Filipin and FITC-conjugated lectin from Ulex Europaeus Agglutinin I (FITC-UEAI). Images were taken on a confocal microscope (SP5, Leica). UV signal (Filipin) was immediately recorded after FITC-UEAI was used to find appropriate z-stack/cellular context.

## Total cholesterol extraction and quantification

Total lipids extracted in 2:1 chloroform methanol. The solution was dried under nitrogen gas. Cholesterol was quantified according to the kit protocol (abcam).

## Lipidomics

Mass spectrometry-based lipid analysis was performed by Lipotype GmbH (Dresden, Germany) as described (*Sampaio et al., 2011*). Lipids were extracted using a two-step chloroform/methanol procedure (*Ejsing et al., 2009*). Samples were spiked with internal lipid standard mixture containing: cardiolipin 14:0/14:0/14:0/14:0 (CL), ceramide 18:1;2/17:0 (Cer), diacylglycerol 17:0/17:0 (DAG), hexosylceramide 18:1;2/12:0 (HexCer), lyso-phosphatidate 17:0 (LPA), lyso-phosphatidylcholine 12:0 (LPC), lyso-phosphatidylethanolamine 17:1 (LPE), lyso-phosphatidylglycerol 17:1 (LPG), lyso-phosphatidylinositol 17:1 (LPI), lyso-phosphatidylserine 17:1 (LPS), phosphatidate 17:0/17:0 (PA), phosphatidylcholine 17:0/17:0 (PC), phosphatidylethanolamine 17:0/17:0 (PE), phosphatidylglycerol 17:0/17:0 (PG), phosphatidylinositol 16:0/16:0 (PI), phosphatidylserine 17:0/17:0 (PS), cholesterol

ester 20:0 (CE), sphingomyelin 18:1;2/12:0;0 (SM), sulfatide d18:1;2/12:0;0 (Sulf), triacylglycerol 17:0/17:0/17:0 (TAG) and cholesterol D6 (Chol). After extraction, the organic phase was transferred to an infusion plate and dried in a speed vacuum concentrator. 1st step dry extract was re-suspended in 7.5 mM ammonium acetate in chloroform/methanol/propanol (1:2:4, V:V:V) and 2nd step dry extract in 33% ethanol solution of methylamine in chloroform/methanol (0.003:5:1; V:V:V). All liquid handling steps were performed using Hamilton Robotics STARlet robotic platform with the Anti Droplet Control feature for organic solvents pipetting. Samples were analyzed by direct infusion on a QExactive mass spectrometer (Thermo Scientific) equipped with a TriVersa NanoMate ion source (Advion Biosciences). Samples were analyzed in both positive and negative ion modes with a resolution of Rm/z=200 = 280,000 for MS and Rm/z=200 = 17,500 for MSMS experiments, in a single acquisition. MSMS was triggered by an inclusion list encompassing corresponding MS mass ranges scanned in 1 Da increments (*Surma et al., 2015*). Both MS and MSMS data were combined to monitor CE, DAG and TAG ions as ammonium adducts; PC, PC O-, as acetate adducts; and CL, PA, PE, PE O-, PG, PI, and PS as deprotonated anions. MS only was used to monitor LPA, LPE, LPE O-, LPI, and LPS as deprotonated anions; Cer, HexCer, SM, LP,C and LPC O- as acetate adducts and cholesterol as ammonium adduct of an acetylated derivative (*Liebisch et al., 2006*). Data were analyzed with in-house developed lipid identification software based on LipidXplorer (*Herzog et al., 2011*; *Herzog et al., 2012*). Data post-processing and normalization were performed using an in-house developed data management system. Only lipid identifications with a signal-to-noise ratio >5, and a signal intensity fivefold higher than in corresponding blank samples were considered for further data analysis.

## ALOD4 and OlyA purification

ALOD4 and OlyA expression constructs were generously provided by the lab of Dr. Arun Radhakrishnan. Recombinant His-tagged ALOD4 and OlyA were purified as previously described (*Endapally et al., 2019b*). Briefly, ALOD4 expression was induced with 1 mM IPTG in $OD_{0.5}$ BL21 (DE3) pLysS *E. coli* for 16 hr at 18 °C. Cells were lysed and His-ALOD4 and His-OlyA were isolated by nickel purification followed by size exclusion chromatography (HisTrap-HP Ni column, Tricorn 10/300 Superdex 200 gel filtration column; FPLC AKTA, GE Healthcare). Protein-rich fractions were pooled and concentration was measured using a NanoDrop instrument.

## ALOD4 fluorescent labeling

20 nmol ALOD4 was combined with 200 nm AlexaFluor maleimide (ThermoFisher) in 50 mM Tris-HCl, 1 mM TCEP, 150 mM NaCl pH 7.5 and incubated at 4 °C for 16 hr. The reaction was quenched using 10 mM DTT. Unbound fluorescent label and DTT were removed by dialysis (EMD Millipore).

## ALOD4 binding and western blot analysis

At time of collection, HUVEC were washed three times for 5 min in PBS with $Ca^{2+}$ and $Mg^{2+}$ containing 0.2% (wt/vol) BSA. Cells were then incubated with 3 µM ALOD4 in basal EBM2 media containing 0.2% (wt/vol) BSA for 1 hr at 4 °C. The unbound proteins were removed by washing three times with PBS with $Ca^{2+}$ and $Mg^{2+}$ for 5 min each. Cells were then lysed and prepared for SDS-PAGE and immunoblotting. ALOD4 was probed on nitrocellulose gels using anti-6X His (abcam) antibody at 15 kDa. A similar method was used for OlyA binding.

## ALOD4 in-cell western blot analysis

Cells were cultured onto 96 wells and ALOD4 binding was performed as mentioned above up until lysis. Cells were directly incubated with DyLight680-conjugated anti-His antibody (Thermofisher), washed, and 700 nm fluorescence was recorded directly on Odyssey CLx platform (LICOR Biosciences).

## ALOD4 flow cytometry analysis

Cells were suspended in PBS with $Ca^{2+}$ and $Mg^{2+}$ containing 2% FBS and washed three times. Binding was with 3 µM ALOD4-647 for 1 hr at 4 °C. Cells were then washed three times with PBS with $Ca^{2+}$ and $Mg^{2+}$ containing 2% FBS and mean fluorescence intensity per cell was measured by LSRII (BD Biosciences) flow cytometer the same day of the assay.

## Animal studies

All animals were handed according to approved institutional animal care and use committee (IACUC) protocols (#07919–2020) of Yale University. At 10 weeks of age, male C57BL/6 J mice (JAX, #000664) were injected with 15 mg/kg lipopolysaccharide (LPS) from *E. coli* O111:B4 intraperitoneally (Sigma). Six hr later, blood was collected for lipid and cytokine analysis. Mice were perfused with PBS and lungs were processed for flow cytometry analysis. Briefly, lung cells were brought to a single-cell suspension via collagenase incubation and then stained for flow cytometry at a concentration of $5 \times 10^6$ cells/mL with Cd31 (Biolegend) and 3 µM ALOD4-47.

## Statistics

Statistical differences were measured with an unpaired two-sided Student's t-test or ANOVA with listed correction for multiple corrections. A value of $p < 0.05$ was considered statistically significant. 'n' within figure legends involving HUVEC denotes number of donors used for the respective experiment. Data analysis was performed with GraphPad Prism software (GraphPad, San Diego, CA).

## Cell Lines

The HUVECs used in this study were primary isolates of ECs collected from donor umbilical cords through Yale School of Medicine's Vascular Biology and Therapeutic core as previously described (*Ewenstein et al., 1987*). No transformation were performed and they were used at low passage (p4).

## Acknowledgements

This work was supported by NIH grant R35HL139945, RO1DK125492, PO1 HL1070205 to WCS and a Supplement to R35HL139945 to JWF and K01DK124441 to NEB. The expression plasmids for ALOD4 and OlyA were graciously provided by Dr. Arun Radhakrishnan.

## Additional information

### Funding

| Funder | Grant reference number | Author |
| --- | --- | --- |
| National Institute of Health and Medical Research | | William C Sessa |

The funders had no role in study design, data collection and interpretation, or the decision to submit the work for publication.

### Author contributions

Joseph Wayne M Fowler, Conceptualization, Data curation, Formal analysis, Visualization, Methodology, Writing - original draft, Project administration, Writing - review and editing; Rong Zhang, Resources, Data curation, Protein Purification; Bo Tao, Validation, Investigation; Nabil E Boutagy, Validation, Writing - review and editing; William C Sessa, Conceptualization, Resources, Supervision, Writing - original draft, Project administration, Writing - review and editing

### Author ORCIDs

Joseph Wayne M Fowler http://orcid.org/0000-0001-5679-2236
William C Sessa http://orcid.org/0000-0001-5759-1938

### Ethics

All animals were handed according to approved institutional animal care and use committee (IACUC) protocols (#07919-2020) of Yale University.

### Decision letter and Author response

Decision letter https://doi.org/10.7554/eLife.79529.sa1
Author response https://doi.org/10.7554/eLife.79529.sa2

## Additional files

### Supplementary files

• Supplementary file 1. RNA-seq normalized counts and lipidomics. (RNA-seq) HUVEC were treated with TNFα for 0, 4, and 10 hr and with or without siRNA targeting *RELA*. (Lipidomics). HUVEC were treated with TNFα for 0, 4, and 10 hr. Data represented as molar percentage of lipid

• MDAR checklist

### Data availability

Sequencing data have been deposited in GEO under accession code GSE201466. All data generated or analysed during this study are included in the manuscript and supporting file; Source Data files have been provided for all figures.

The following dataset was generated:

| Author(s) | Year | Dataset title | Dataset URL | Database and Identifier |
|---|---|---|---|---|
| Fowler JW, Zhang R, Tao B, Boutagy NE, Sessa WC | 2022 | Inflammatory stress signaling via NF-kB alters accessible cholesterol to upregulate SREBP2 transcriptional activity in endothelial cells | https://www.ncbi.nlm.nih.gov/geo/query/acc.cgi?acc=GSE201466 | NCBI Gene Expression Omnibus, GSE201466 |

The following previously published dataset was used:

| Author(s) | Year | Dataset title | Dataset URL | Database and Identifier |
|---|---|---|---|---|
| Romanoski CE, Hogan NT | 2017 | Genome-wide map of HAEC chromatin landscape under resting and TNFa, IL1b, and OxPAPC stimulation, with corresponding transcription factor binding and RNA expression | https://www.ncbi.nlm.nih.gov/geo/query/acc.cgi?acc=GSE89970 | NCBI Gene Expression Omnibus, GSE89970 |

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

# Appendix 1

## Appendix 1—key resources table

| Reagent type (species) or resource | Designation | Source or reference | Identifiers | Additional information |
|---|---|---|---|---|
| antibody | Anti-His Tag Dylight 680 (Mouse monoclonal) | ThermoFisher Scientific | MA1-21315-D680 | In-cell Western (1:1,000) |
| antibody | Anti-6x His (Mouse monoclonal) | abcam | ab18184 | WB (1:1,000) |
| antibody | Anti-ABCG1 (Rabbit monoclonal) | abcam | ab52617 | WB (1:1,000) |
| antibody | Anti-mmCd31-BV605 (Rat monoclonal) | BioLegend | 102427 | FACS (1:200) |
| antibody | Anti-GAPDH (Rabbit monoclonal) | Cell Signalling | 2118 S | WB (1:2,000) |
| antibody | Anti-HSP90 (Mouse monoclonal) | Santa Cruz | sc-13119 | WB (1:2,000) |
| antibody | Anti-ICAM1 (Rabbit monoclonal) | Cell Signaling | 4915 S | WB (1:1,000) |
| antibody | Anti-JNK1 (Mouse monoclonal) | Cell Signaling | 3708 S | WB (1:1,000) |
| antibody | Anti-LC3b (Rabbit monoclonal) | Cell Signaling | 2775 S | WB (1:1,000) |
| antibody | Anti-LDLR (Rabbit monoclonal) | abcam | ab52818 | WB (1:1,000) |
| antibody | Anti-p-JNK1 (Rabbit monoclonal) | Cell Signaling | 9261 S | WB (1:1,000) |
| antibody | Anti-p-p38 (Rabbit monoclonal) | Cell Signaling | 9211 S | WB (1:1,000) |
| antibody | Anti-p38 (Rabbit monoclonal) | Cell Signaling | 9212 S | WB (1:1,000) |
| antibody | Anti-P65/RELA (Rabbit monoclonal) | Cell Signaling | 8242 S | WB (1:1,000) |
| antibody | Anti-SREBP1a (Mouse monoclonal) | Santa Cruz | sc-13551 | WB (1:1,000) |
| antibody | Anti-SREBP2 (Mouse monoclonal) | BD Biosciences | 557037 | WB (1:1,000) |
| antibody | Anti-STARD10 (Rabbit polyclonal) | Thermofisher Scientific | PA5-36947 | WB (1:1,000) |
| antibody | Anti-VCAM1 (Mouse monoclonal) | Santa Cruz | sc-13160 | WB (1:1,000) |
| sequence-based reagent | siRNA: *RELA* | Thermofisher Scientific | s11914 | Silencer Select |
| sequence-based reagent | siRNA: *SREBF2* | Thermofisher Scientific | s27 | Silencer Select |
| sequence-based reagent | siRNA: *HMGCR* | Thermofisher Scientific | 110740 | Silencer |
| sequence-based reagent | siRNA: *SCAP* | Thermofisher Scientific | s695 | Silencer Select |
| sequence-based reagent | siRNA: *STARD10* #1 | Thermofisher Scientific | s21244 | Silencer Select |
| sequence-based reagent | siRNA: *STARD10* #2 | Thermofisher Scientific | s21243 | Silencer Select |
| sequence-based reagent | siRNA: *ABCG1* #1 | Thermofisher Scientific | s18482 | Silencer Select |
| sequence-based reagent | siRNA: *ABCG1* #2 | Thermofisher Scientific | S18484 | Silencer Select |

*Appendix 1 Continued on next page*

*Appendix 1 Continued*

| Reagent type (species) or resource | Designation | Source or reference | Identifiers | Additional information |
|---|---|---|---|---|
| sequence-based reagent | hs*ACTB*_F | This Paper | qRT-PCR Primers | AGCACTGTGTTGGCGTACAG |
| sequence-based reagent | hs*ACTB*_R | This Paper | qRT-PCR Primers | GGACTTCGAGCAAGAGATGG |
| sequence-based reagent | hs*LDLR*_F | This Paper | qRT-PCR Primers | TCTGCAACATGGCTAGAGACT |
| sequence-based reagent | hs*LDLR*_R | This Paper | qRT-PCR Primers | TCCAAGCATTCGTTGGTCCC |
| sequence-based reagent | hs*HMGCS1*_F | This Paper | qRT-PCR Primers | CAAAAAGATCCATGCCCAGT |
| sequence-based reagent | hs*HMGCS1*_R | This Paper | qRT-PCR Primers | AAAGGCTTCCAGGCCACTAT |
| sequence-based reagent | hs*HMGCR*_F | This Paper | qRT-PCR Primers | TGATTGACCTTTCCAGAGCAAG |
| sequence-based reagent | hs*INSIG1*_F | This Paper | qRT-PCR Primers | CTAAAATTGCCATTCCACGAGC |
| sequence-based reagent | hs*INSIG1*_R | This Paper | qRT-PCR Primers | GCACTGCATTAAACGTGTGG |
| sequence-based reagent | hs*SREBF2*_F | This Paper | qRT-PCR Primers | TAAAGGAGAGGCACAGGA |
| sequence-based reagent | hs*SREBF2*_R | This Paper | qRT-PCR Primers | AGGAGAACATGGTGCTGA |
| sequence-based reagent | hs*ICAM1*_F | This Paper | qRT-PCR Primers | GTGGTAGCAGCCGCAGTC |
| sequence-based reagent | hs*ICAM1*_R | This Paper | qRT-PCR Primers | GGCTTGTGTGTTCGGTTTCA |
| sequence-based reagent | hs*CXCL1*_F | This Paper | qRT-PCR Primers | AGGGAATTCACCCCAAGAAC |
| sequence-based reagent | hs*CXCL1*_R | This Paper | qRT-PCR Primers | TGGATTTGTCACTGTTCAGCA |
| sequence-based reagent | hs*SELE*_F | This Paper | qRT-PCR Primers | ACCTCCACGGAAGCTATGACT |
| sequence-based reagent | hs*SELE*_R | This Paper | qRT-PCR Primers | CAGACCCACACATTGTTGACTT |
| sequence-based reagent | hs*SCAP*_F | This Paper | qRT-PCR Primers | CGCAAACAAGGAGAGCCTAC |
| sequence-based reagent | hs*SCAP*_R | This Paper | qRT-PCR Primers | TGTCTCTCAGCACGTGGTTC |
| sequence-based reagent | hs*STARD10*_F | This Paper | qRT-PCR Primers | GAAAGACTTGGTCCGAGCTG |
| sequence-based reagent | hs*STARD10*_R | This Paper | qRT-PCR Primers | TTCCACTCGGGGTACTTGAG |
| chemical compound, drug | 25-hydroxycholesterol | Sigma Aldrich | H1015 | |
| chemical compound, drug | Actinomycin D | ThermoFisher | 11805017 | |
| chemical compound, drug | BAY 117082 | Sigma Aldrich | B556-10MG | |
| chemical compound, drug | Cholesterol, 1,2-3H(N) | Perkin Elmer | NET139250UC | |
| chemical compound, drug | Choroquine (CQ) | Sigma Aldrich | C6628 | |

*Appendix 1 Continued on next page*

*Appendix 1 Continued*

| Reagent type (species) or resource | Designation | Source or reference | Identifiers | Additional information |
|---|---|---|---|---|
| chemical compound, drug | DiI LDL | Kalen Biomedical | 770230 | |
| chemical compound, drug | EGM2 | Lonza | CC-3162 | |
| chemical compound, drug | Fatostatin | Cayman | 13562 | |
| chemical compound, drug | Filipin | Cayman | 70440 | |
| chemical compound, drug | FITC-UEAI | ThermoFisher | L32476 | |
| chemical compound, drug | Lipopolysaccharide from *E. coli* O111:B4 | Sigma Aldrich | L2630 | |
| chemical compound, drug | Lipoprotein Depleted Serum (LPDS) | Kalen Biomedical | 880100 | |
| chemical compound, drug | MβCD | Sigma Aldrich | C4555 | |
| chemical compound, drug | MβCD-Cholesterol | Sigma Aldrich | C4951 | |
| chemical compound, drug | Native LDL | Kalen Biomedical | 770200 | |
| chemical compound, drug | PF-429242 | Sigma Aldrich | SML0667 | |
| chemical compound, drug | rhIL1β | RD Systems | 201-LB-010/CF | |
| chemical compound, drug | rhTNFα | RD Systems | 210-TA-020/CF | |
| chemical compound, drug | Sandoz 58–035 | Sigma Aldrich | S9318-25mg | |
| chemical compound, drug | Sphingomyelinase | Sigma Aldrich | S8633 | |
| chemical compound, drug | T0901317 | Sigma Aldrich | T2320 | |
| chemical compound, drug | Triacin C | RD Systems | 2472 | |
| chemical compound, drug | U18666A | Sigma Aldrich | U3633 | |
| commercial assay or kit | Cholesterol/Cholesteryl Ester Assay Kit | Abcam | Ab65359 | |
| software, algorithm | Partek Flow | Partek | https://www.partek.com/partek-flow/ | |
| software, algorithm | Ingenuity Pathway Analssis | Qiagen | https://digitalinsights.qiagen.com/products-overview/discovery-insights-portfolio/analysis-and-visualization/qiagen-ipa/ | |

