## [Editor Report]

This is a fundamental contribution to linking inflammation to cholesterol metabolism in endothelial cells. The strength of evidence was compelling and the uncovering of the molecular mechanisms underlying the pathway was a significant addition to the overall value of the study.

---

## [Decision Letter]

**Decision letter after peer review:**

Thank you for submitting your article "Inflammatory stress signaling via NF-κB alters accessible cholesterol to upregulate SREBP2 transcriptional activity in endothelial cells" for consideration by *eLife*. Your article has been reviewed by 2 peer reviewers, and the evaluation has been overseen by a Reviewing Editor and Mone Zaidi as the Senior Editor. The reviewers have opted to remain anonymous.

Essential revisions:

You will see that one reviewer (#1) had a number of clarifications and extensions of your results that are not expected to involve significant experimental work. Reviewer #3 and I (in discussions after the reviews were received) agreed with the editorial changes requested by reviewer #1. Thus, please address all of them. In terms of data clarification/extension, please address reviewer 1's requests:

1) Figure 3b: quantification of full length vs cleaved SREBP2.

2) Figure 6: RELA occupancy of other potential RELA-dependent genes.

3) Western blots of STARD10 after TNF treatment or RELA knockdown.

*Reviewer #1 (Recommendations for the authors):*

Overall this is a very thorough set of experiments and a well-written manuscript. There are a few areas that could be strengthened.

Figure 2 b: The rapid activation of NF-Kappa-B genes by TNF is likely a result of RNA polymerase being poised at the promoter for rapid response (see PMID: 19820169). It might be useful to mention this in the results or discussion as an explanation for the lag in the induction of the SREBP2 genes relative to the NF-Kappa-B targets.

Figure 3b: The authors should quantitate the full length vs cleaved SREBP2 upon Actinomycin D vs vehicle treatment. It's difficult to glean the main point from this figure since there appears to be less of the unprocessed form of SREBP upon Actinomycin D treatment.

Figure 6a: There were several other lipid mediator genes (other than STARD10) that are RELA dependent. It would be useful to see if the other genes also show RELA occupancy, which could help determine specificity as it is likely that STARD10 is not the whole story. It would also be useful to discuss one or two of these other factors and how they might (or might not) impact cholesterol accessibility.

There were no western blots of STARD10 upon TNF stimulation or RELA knockdown. It would be important to know whether the STARD10 protein abundance is reduced by the siRNA as much as the mRNA since the effect of STARD10 depletion on SREBP2 cleavage is modest (Figure 6D).

In figure 6c, there needs to be a scale on the ChIP seq data of RELA with STARD10 since the tracks without this are not very meaningful. Also, the authors might not want to use the term "strong" binding of RELA to the STARD10 promoter (strong relative to what?) in the Results section but rather state that there is an increase in occupancy of RELA upon TNF and IL1B stimulation.

This reviewer has never liked the use of the term "master" regulator for transcription factors. We know SREBP is a key regulator of cholesterol biosynthesis gene expression, but this terminology is outdated and potentially offensive to some readers. I would suggest removing it from the manuscript.

---

## [Author Response]

Reviewer #1 (Recommendations for the authors):Overall this is a very thorough set of experiments and a well-written manuscript. There are a few areas that could be strengthened.Figure 2 b: The rapid activation of NF-Kappa-B genes by TNF is likely a result of RNA polymerase being poised at the promoter for rapid response (see PMID: 19820169). It might be useful to mention this in the results or discussion as an explanation for the lag in the induction of the SREBP2 genes relative to the NF-Kappa-B targets.

We have added reference to RNA pol II priming of rapid NF-κB response gene transcription in the Results section.

Figure 3b: The authors should quantitate the full length vs cleaved SREBP2 upon Actinomycin D vs vehicle treatment. It's difficult to glean the main point from this figure since there appears to be less of the unprocessed form of SREBP upon Actinomycin D treatment.

We have quantified the SREBP2 precursor (P) and cleaved product (C) normalized by a loading control , Hsp90, for the actinomycin D experiments and is found in panel C.

Figure 6a: There were several other lipid mediator genes (other than STARD10) that are RELA dependent. It would be useful to see if the other genes also show RELA occupancy, which could help determine specificity as it is likely that STARD10 is not the whole story. It would also be useful to discuss one or two of these other factors and how they might (or might not) impact cholesterol accessibility.

We have added to Figure Supplement 2 showing RELA ChIP-seq analysis for TNF-inducible lipid mediators.

We have also added data to Figure Supplement 3 where we explored ABCG1 as RELA-inducible target upstream of cholesterol regulation. Two independent siRNAs did not impact cholesterol accessibility.

There were no western blots of STARD10 upon TNF stimulation or RELA knockdown. It would be important to know whether the STARD10 protein abundance is reduced by the siRNA as much as the mRNA since the effect of STARD10 depletion on SREBP2 cleavage is modest (Figure 6D).

After testing multiple Abs, we now have found one that works for STARD10 and have added this to Figure 6C.

In figure 6c, there needs to be a scale on the ChIP seq data of RELA with STARD10 since the tracks without this are not very meaningful. Also, the authors might not want to use the term "strong" binding of RELA to the STARD10 promoter (strong relative to what?) in the Results section but rather state that there is an increase in occupancy of RELA upon TNF and IL1B stimulation.

Figure 6b: Scale bards are added to ChIP-seq

Edited terminology in discussing RELA ChIP seq data of STARD10 promoter

This reviewer has never liked the use of the term "master" regulator for transcription factors. We know SREBP is a key regulator of cholesterol biosynthesis gene expression, but this terminology is outdated and potentially offensive to some readers. I would suggest removing it from the manuscript.

We have removed “master” terminology